# Oral Health, Health Service Utilization, and Age at Arrival to the U.S. among Safety Net Patients

**DOI:** 10.3390/ijerph19031477

**Published:** 2022-01-28

**Authors:** Sarah E. Raskin, R. Rasnick, Tatiana Kohlmann, Martin Zanin, Julie Bilodeau, Aderonke Akinkugbe

**Affiliations:** 1L. Douglas Wilder School of Government and Public Affairs, Virginia Commonwealth University, 1001 W. Franklin St., Richmond, VA 23284, USA; 2VCU Institute for Inclusion, Inquiry and Innovation (iCubed) Oral Health Core, Virginia Commonwealth University, 912 West Grace St., Richmond, VA 23284, USA; aaakinkugbe@vcu.edu; 3Department of Biostatistics, VCU School of Medicine, Virginia Commonwealth University, 830 East Main Street, Richmond, VA 23219, USA; rasnickr@vcu.edu; 4Department of Health Behavior & Policy, VCU School of Medicine, Virginia Commonwealth University, 830 East Main Street, Richmond, VA 23219, USA; Tatiana.Kohlmann@vcuhealth.org; 5CrossOver Healthcare Ministry, 8600 Quioccasin Rd., Richmond, VA 23229, USA; mzanin@crossoverministry.org (M.Z.); jbilodeau@crossoverministry.org (J.B.); 6Department of Dental Public Health and Policy, VCU School of Dentistry, Virginia Commonwealth University, 1101 East Leigh Street, Richmond, VA 23298, USA

**Keywords:** lifecourse, immigration, migration, oral health, dental care, health service utilization, health care safety net, oral health related quality of life (OHRQoL)

## Abstract

Background: Immigrants’ oral health disparities have not been adequately investigated using a lifecourse approach, which investigates the cumulative effects of risk and protective exposures among other considerations. Methods: We examined self-reported oral health outcomes and health care appointment outcomes among a sample of patients enrolled at a federally qualified health center in Richmond Virginia (N = 327) who were categorized into three groups by approximate age at arrival to the U.S. Results: Study participants who arrived to the U.S. prior to age 18 had better retention of natural dentition, better oral health related quality of life, and a lower proportion of dental appointments to address pain than those who arrived after age 18 or were born in the U.S. Conclusions: Im/migrants’ differentiated oral health outcomes by age at arrival to the U.S. suggest the relevance of lifecourse factors, for example the cumulative effects of risk and protective exposures, and confirm the merits of lifecourse studies of im/migrants’ oral health.

## 1. Introduction

Oral health is an essential part of overall health. Clinical and self-reported oral health outcomes, including periodontal (gum) disease, loss of permanent dentition, oral pain, oral health related quality of life (OHRQoL), and dental caries (cavities)—the most common disease worldwide—develop from a variety of hereditary, behavioral, sociopolitical, corporate, and environmental factors [1,2,3]. Notably, these factors interact in complex, dynamic, and often ongoing ways that cumulatively influence individuals’ and populations’ oral health over time.

Lifecourse theory strengthens causal explanations of health outcomes, in particular population-level inequities in preventable or treatable disease and associated suffering [3,4,5,6]. Oral health scholars increasingly utilize lifecourse theory to identify root causes across the lifespan—from gestation through aging, and at pivotal biopsychosocial points throughout—that increase susceptibility to otherwise preventable oral disease in the long term. Causes of preventable oral diseases that have been documented using a lifecourse approach include gestational environment, “early programming” of inadequate home hygiene, psychosocial stress, and exclusions from routine oral care where preventive services are delivered and early stage disease identified and resolved [3,4,6,7].

Among the lifecourse concepts most commonly investigated in oral health studies is the effects of childhood home socioeconomic status on individuals’ oral health throughout childhood, adolescence, and adulthood. Parents’ income and educational attainment predict progenies’ oral pain, periodontitis, loss of permanent teeth, oral-related psychosocial distress, OHRQoL and dental attendance not only into midlife but also into advanced age, as do intergenerational economic mobility (the likelihood of children’s lifetime economic stability exceeding their parents’) and conditions of employment including workforce security [8,9,10,11,12,13]. These studies evince lifecourse theories of the cumulative impact and persisting pathways of interacting risk or protective exposures [3,4,5,6] that compound other well-demonstrated risk and protective exposures such as sugar intake, exposure to fluoride through public water systems, and home hygiene and routine oral care attendance [9,14,15]. By contrast, other factors remain critically underexamined in lifecourse studies of oral health, in particular sociopolitical factors [5].

Immigration is increasingly understood as a sociopolitical determinant of health in general, and specifically as phenomenon through which, following lifecourse theory, prior and ongoing interacting risk and protective exposures subsequently influence health [16,17,18,19]. Immigration-specific factors that can affect health outcomes long term include im/migration context (elective, necessitated, or forced); living conditions prior to, during, and subsequent to arrival including housing, health care, and nutritional security and quality; and psychosocial experiences throughout, for example the stress of family disruption or discrimination. For example, younger age-at-migration or being born in the U.S. is associated with obesity among heterogeneous Latinx/Hispanic-Americans, as well as with depression among older Mexican-Americans [20,21,22]. While age-at-migration is conventionally (and increasingly controversially) understood as a proxy measure for acculturation, a lifecourse understanding of age with regard to migration dynamics can, through a lifecourse approach, suggest duration of exposures to risks for which cumulative effects are meaningful, for example inadequate water, sanitation resources, or nutrition in communities of origin that experienced disinvestments by the state, or harmful “conveniences” of higher income countries such as excess processed foods and other addictive substances [16,23].

Minoritized, marginalized, and oppressed populations groups bear a disproportionate excess of unresolved oral diseases, including the U.S.’s over 86 million foreign-born or “first generation” children of foreign-born residents and the sub-groups therein [24,25]. For example, non-US-born residents are approximately half as likely to have had a past year dental visit as their U.S.-born counterparts, and these visits were more likely to be for tooth extraction and less likely for comprehensive dental care or a full examination [26,27]. Latinx/Hispanic adults who report need of a dental appointment—a validated measure of clinical diagnoses—were less likely to have had a past year appointment one regardless of country of birth [28,29]. Relationships between immigration and oral health have seldom been considered using a lifecourse approach. Similarly, oral conditions have rarely been examined in lifecourse studies of immigration and health. This study addresses these knowledge gaps. The purpose of this study was to determine the relationships between age at arrival to the U.S. and a number of oral health outcomes including self-reported measures and health services utilization measures, both oral health utilization by service type and as compared with medical and mental health services.

## 2. Materials and Methods

Data analyzed in this study were drawn from a mixed methods cross-sectional study of oral health and dental utilization experience among patients who obtained care between 1 August 2018 and 19 March 2021 at CrossOver Healthcare Ministry, a federally qualified health center (FQHC) in Richmond, Virginia. This community engaged study was developed through collaboration between the university-based research team and FQHC administrators.

### 2.1. Setting, Study Population and Participants

CrossOver Healthcare Ministry (CrossOver) is a safety net clinic that delivers comprehensive medical services and an expanded array of pharmacy, vision, and dental services to adult and child patients. In 2019, CrossOver served 6679 unique patients with a total of 15,819 primary medical visits, 3666 dental visits, and 2061 mental health visits. CrossOver’s clinical services are primarily delivered by volunteers, who in 2019 provided a total of 13,957 h, or the equivalent of seven full-time employee providers. CrossOver operated continuously throughout the COVID-19 pandemic. Three-quarters of CrossOver’s patients identify a language other than English as their first language; they hail from 118 countries.

We enrolled study participants using a simple random sampling strategy, initially recruiting patients in-person in the waiting areas of CrossOver’s two sites with recruitment lists provided by staff then, following the introduction of COVID-19 in the region in Spring 2020, using CrossOver’s text-based communications system. Eligibility criteria included being a current patient at CrossOver, being 18 years of age or older, and speaking English or Spanish. Inclusion criteria was the same as eligibility criteria, with the addition of consenting to participate in the study. Exclusion criteria was not being a current patient at CrossOver, not being 18 years of age or older, not speaking English or Spanish, or not consenting to participate in the study. The sample size assumed a dental appointment cancellation/no-show prevalence of 20%, a 5% margin of error, and a 50% response distribution to maximize sampling size.

### 2.2. Data Source

We collected data from N = 327 adults ages 18 to 80. Participants completed an original survey instrument (see Appendix A) and the Oral Health Impact Profile-14 (OHIP-14), an instrument used to measure oral health related quality of life (OHRQoL) that has been translated into and validated in numerous languages and among many populations globally. Participants who enrolled in the study in-person (prior to March 2020) completed paper forms in English or Spanish in a private location at the clinic, either completing the forms by hand or, if preferring to respond verbally likely due to a literacy limitation, by reporting their answers to a bilingual/bicultural research assistant who read them the form. Participants who enrolled in the study via telecommunications following COVID-19-necessitated changes (after August 2020) reported their responses to a bilingual/bicultural research assistant who read them the survey in English or Spanish by phone. Self-report study data were managed using REDCap electronic data capture tools hosted at Virginia Commonwealth University [30,31]. Primary data sources were matched with patient medical and appointment information extracted from their electronic health record (EHR).

All research participants consented to participate in the study prior to data collection. This study was approved on 20 May 2019 by the Virginia Commonwealth University (VCU) Institutional Review Board (IRB), Protocol #HM20014861.

### 2.3. Outcomes

#### 2.3.1. Self-Reported Oral Health Outcomes

We documented participants’ number of missing permanent teeth by asking, “How many of your permanent teeth have you lost, not including 3rd molars?” We categorized the number of missing teeth as none, 1–8, and more than 8. We documented participants’ self-reported oral health status by asking “How would you rate the status of your dental health?” We categorized responses as poor/fair and good/very good/excellent. We documented participants’ last dental visit by asking “When was the last time you visited a dentist?” We categorized responses into 4 groups: within 1 year, within 2 years, within 5 years, and more than 5 or never.

We gathered participant OHRQoL data using the OHIP-14. The OHIP-14 uses a Likert-scale where 1 = Never, 2 = Hardly Ever, 3 = Occasionally, 4 = Fairly Often, and 5 = Very Often. Importantly, lower scores indicate higher OHRQoL and, conversely, higher scores indicate worse OHRQoL. Summed scores ranged from 14 to 70. We classified scores into tertiles by using the 33rd percentile and 66th percentile to account for the asymmetry of summing: good OHRQoL where OHIP < 17, moderate OHRQoL where 17 < OHIP ≤ 26, and poor OHRQoL where OHIP > 26 OHIP score during data analysis.

#### 2.3.2. Health Care Appointment Outcomes

We documented participants’ health care visit outcomes during the study period, including dental visits, medical visits, and mental health. We extracted from EHR records the following outcomes for each visit type, resulting in 9 variables: completed visits, re-scheduled visits, and cancelled visits/no shows. We classified the outcome as none or at least once for each visit type.

We gathered dental appointment types from EHR records and separated them into 4 categories: pain, preventative, restoration, and extractions. Pain includes: pain evaluations, all appointments that include pain, prescriptions, and abscess. Preventative was: scaling/curettage/root planting, cleaning, dental exams/evals, consultations, diabetic appointments, oral hygiene education, enameloplasty Sealant Technique, evaluations on dentures/fillings/lesions/sores, fixing dentures, and all non-specified appointments performed by a dental hygienist. Restorations comprised of fillings, restorations, bridge falling off, endodontic, recementing crowns, and fixing upper dentures. Finally, extractions consisted of: extractions, broken/cracked teeth, dry sockets, and possible extractions.

### 2.4. Measures

Gender was categorized as Male or Female. Language was recorded as English, Spanish, or other/multilingual. Marital status was recorded as married or not. Education was reported as less than high school, high school, some college, and college or more. Alcohol consumption was recorded as either yes or no. Smoking was recorded as never, former, or current. Income was categorized as $0-$10,000, $10,000–20,000, and $20,000+.

### 2.5. Main Predictor

In order to estimate age at arrival to the U.S. We did not differentiate between age at first arrival, age at most recent arrival, or any other variations. CrossOver’s requirement of an annual medical examination for ongoing enrollment indicates continuity of U.S. residence. (Approximate Age at Arrival to the U.S., or AAAU) we needed to first estimate time in the US. Participants self-reported duration of U.S. residence with categories: 0–5 years, 5–10 years, at least 10 years, or born in the US. To calculate time in the U.S.: if they had arrived 0–5 years ago a value of 2.5 was used, if 5–10 a value of 7.5, if at least 10 years the midpoint of 10 and their age was used, and if born in the U.S. their age was their time in the U.S. The AAAU was obtained by the difference in age and time in the US. Since most of the participants were older when arriving in the U.S., AAAU was categorized as less than 18 years of age, at least 18 years of age, and U.S. born (USB). It is labeled for reporting purposes as follows:AAAU < 18;AAAU ≥ 18;USB.

### 2.6. Statistical Analysis

We calculated overall frequency and percentages of socio-demographic variables for all study participants and according to AAAU categories using Chi-square or Fisher’s exact test to assess differences in socio-demographic factors for the AAAU groups.

To summarize the dental specific questions and appointment type/services questions, we also estimated the N (%) overall and according to AAAU groups and summarized the OHIP-14 scores with the medians and interquartile and assessed differences in median OHI-14 scores according to AAAU groups using the Kruskal–Wallis test.

We performed logistic regression reporting odds ratios and 95% confidence intervals to assess the associations between AAAU and current dental outcomes including tooth loss (<8 vs. ≥8), self-reported dental health (poor/fair vs. good/very good/excellent), last dental visit (<1 year vs. ≥1 year), median OHIP-14 score, and each category of appointment outcome by health service type, i.e., completed, incomplete and rescheduled dental, medical and mental health appointment categorized as yes or no. AAAU < 18 served as the reference group against which we tested US-born and AAAU ≥ 18. Final logistic regression models were adjusted for age and gender.

Lastly, we examined the kinds of dental appointments (pain, preventative, restorative, extractions) that were completed, incomplete or rescheduled at least once according to see if there are differences according to approximate age at migration to the U.S. Because it is possible for the same individual to have complete, incomplete or rescheduled appointments based on if it is a preventative, restorative, pain or extraction visit, the estimates produced for this analysis accounts for this overlap.

We conducted all analyses using SAS v.9.4 (SAS Institute Cary, NC and reporting adhered to the STROBE guidelines for observational studies.

## 3. Results

### 3.1. Descriptive Data

#### Socio-Demographic Data

We enrolled a total of 327 participants into the study (Table 1) of whom the majority were female (71%) and prefer to speak Spanish (68%). Approximately half of participants have lived in the U.S. for more than a decade (52%), whereas a third have lived in the U.S. for less than ten years (33%). Most participants are unmarried (62%) and approximately 15% are at least 60 years old. The AAAU < 18 group had a median current age of 38 years old, the AAAU ≥ 18 s and U.S.-born participants each had a current median age of 50 years Table 1.

The distribution of educational attainment at the time of data collection is statistically significant. The AAAU < 18 s were more likely to have completed less than a high school diploma (66%) with U.S. Born participants least likely (*p* < 0.0001) AAAU ≥ 18 s were most likely to have a college or more educational attainment. More than half neither drink alcohol (68%) nor currently smoke cigarettes (90.6%). US Born participants were significantly more likely to be current alcohol consumers and current smokers. More than one-quarter of participants have diabetes (26%) Table 1.

### 3.2. Outcome Data

#### 3.2.1. Oral Health Data

Most participants rated their dental health as poor/fair (58.2%), with no statistically significant difference among the AAAU groups. Three-quarters of participants were missing at least one permanent tooth, with the majority missing between 1 and 8 teeth (57%). U.S. Born participants were most likely to be have one or more missing teeth (84.1%) and the AAAU < 18 s least likely (60%). Intact dentition was highest among AAAU < 18 s (40.5%) as compared with AAAU ≥ 18 s (21.3%) or USBs (15.9%) Table 2.

The median OHIP-14 score across all participants was 21, indicating a moderate oral health related quality of life with an interquartile range of 16 (good OHRQoL)) to 31 (poor OHRQoL). USBs had a higher median OHIP-14 score (24.5) than the other 2 groups. Nevertheless, the OHIP_14 score for the 3 AAAU groups were all moderate (19) based on the tertile categorization.

More than half of all participants had a dental visit within the last year (55%) though past year dental visits were most common among AAAU < 18 s (66.7%) and least common among USBs (34.1%) Table 2.

#### 3.2.2. Dental, Medical and Mental Health Care Appointment Data

With regard to appointment completion status, medical appointments were most likely to be completed (98%), and also to record incomplete/no-showed (87.3%) as compared to dental and mental health appointments.

With regard to incomplete/no-show appointments, 37% of all participants had incomplete/no-show dental appointments, with all AAAU groups tracking closely. Conversely, 27% had incomplete/no-show mental health appointments, with the proportionately fewest mental health appointment incompletes/no-shows among AAAU < 18 s (17%) and the most incomplete/no-show mental health appointments among USBs (43%). Table 3.

#### 3.2.3. Dental Appointment Outcomes

Preventive appointments were the most common dental appointment types across all appointment statuses and among all AAAU groups (Table 4), comprising at minimum 39% of completed appointment among USB and at maximum 54% of completed appointment among AAAU < 18 s. Preventive appointments were also more commonly cancelled/no-showed or rescheduled than other appointment types, with incomplete preventive appointments ranging from 42% among AAAU < 18 to 60% among AAAU ≥ 18. 

Restorations were the second-most completed appointment types across all approximate age-at-migration groups, and the second-most incomplete or rescheduled appointment types. AAAU < 18 s had the most incomplete restoration appointments (38%), while USBs had the most complete restorative appointments (32%). Extractions appointments were least likely to be completed across all age-at-migration categories, comprising 14.3% of completed appointments or less. Extractions were more commonly incomplete or rescheduled among non-U.S. born participants than among USBs, though the small number of extraction appointments of all appointment statuses among USBs limits conclusivity.

Appointments to address pain were commonly completed among each AAAU group, as compared to being incomplete or rescheduled.

### 3.3. Main Results

AAAU ≥ 18 s and USBs were about three times as likely to have more lost teeth than AAAU < 18 s (Figure 1) in the unadjusted analysis. Upon adjustment for age and gender, the estimate for AAAU ≥ 18 s remained statistically significant but the adjusted estimate for USBs lost statistical significance and was attenuated towards the null. There was no significant association between AAAU groups and self-rated dental health.

Both AAAU ≥ 18 s and USBs were less likely to have had a past year dental visit as compared to their AAAU < 18 counterparts Both AAAU ≥ 18 s and USBs have greater odds of worse OHRQoL scores than their AAAU < 18 counterparts (Figure 1).

AAAU ≥ 18 s and USBs as compared to AAAU < 18 have lower odds of completing a dental appointment. While they also have lower odds of incomplete and rescheduled dental appointments, the associations did not attain statistical significance and thus inconclusive (Figure 2).

## 4. Discussion

We found meaningful differences in self-reported oral health outcomes and health services utilization among immigrant safety net patient groups, by age-at-arrival to the United States. Findings reflect other studies’ observations of differences in health outcomes by age-at-migration. While age-at-migration has commonly and controversially been used as a proxy variable for acculturation [23], lifestage measures such as duration of time in a specific community context may also indicate, using lifecourse theory, cumulative exposure to risk or protective factors that have lasting impacts on health [3,4,5]. This is the first study, to our knowledge, to apply a lifecourse approach to examine immigrant safety net patients’ oral health self-report and service utilization outcomes. Results suggest the merits of conducting more lifecourse research on immigrants’ oral health as well as that of their children, and of incorporating oral health into other lifecourse studies of immigrants’ overall health, especially with regard to conditions with shared risk factors or co-morbid interactions [3,5,14,17].

Study participants who arrived to the U.S. prior to age 18 had better retention of natural dentition than those who arrived after age 18 or were born in the U.S. This finding may reflect the study population’s on-average younger median age (38 years old) as compared with their counterparts’ median age (50 years old). AAAU < 18 s comparatively strong oral health as compared with USBs may also be attributable to the formers’ higher past year completion of a preventive dental visit and lower use of alcohol and tobacco. AAAU < 18 s comparatively strong oral health as compared with AAAU ≥ 18 s may also reflect the former’s earlier exposure to fluoridated water, home hygiene materials, and routine preventive oral services, a hypothesis that the authors will explore in analyses of qualitative data collected for the broader study. AAAU < 18 s superior retention of dentition, an indicator of gum health, differs from other studies of chronic disease by age-at-migration, specifically metabolic disorder indicators which are associated across the lifespan with younger age-at-migration [20,21,22]. Future lifecourse studies of immigrants’ health should investigate conditions that have risk and protective factors, for example sugar intake and microbiome composition as related to oral conditions and metabolic disorders, particularly when initial evidence suggests meaningful contradictions such as those shared here.

Study participants who arrived to the U.S. prior to age 18 also had better oral health related quality of life than those who arrived after age 18 or were born in the U.S, and better self-rated oral health than their U.S.-born counterparts. This finding may be partially explained by their lower proportion of dental pain appointments overall, indicating a correlation between need of dental pain treatment and self-rated oral health or OHRQoL among USBs and AAAU ≥ 18 s. This finding may also indicate psychosocial factors that will be explored through analyses of qualitative data such as participants’ different expectations of their dentition that may change through different contexts, for example community norms or with aging.

The popularity of preventive oral health services completed, incomplete, and rescheduled across all groups was unsurprising, given the comparative volume of preventive dental appointments at CrossOver versus other appointment types. Similarly, the strong medical appointment completion and frequency of incompletion and rescheduling is also unsurprising, given the clinic’s requirement that patients complete an annual medical visit to remain active on its roster.

The strong utilization of medical, dental, and mental health services by participants who arrived to the U.S. after age 18 may reflect a variety of explanations. CrossOver Healthcare Ministry prioritizes dental services for aging populations among others (patients who are pregnant, have diabetes, or are HIV+). While AAAU ≥ 18 s had the same median age as USBs, they also had the highest age range among enrolled participants. Depending on participants’ communities of origin, and pre-migration opportunities to receive health services depending on state dis/investment in health care delivery, the opportunities to receive long-delayed care may drive strong utilization among these participants, as may being in retirement age, versus AAAUs < 18, whose conditions of employment and home responsibilities may inhibit appointment completion due to scheduling and other conflicts.

This study finds that AAAU < 18 s’ oral health outcomes are more favorable than USBs’ across nearly every measure. This set of findings contrasts with existing evidence that immigrants have comparatively poorer oral health outcomes than their US-born counterparts [24,25,26,27,28,29], and merits special consideration. Differences observed may be attributable to a variety of factors. Income is one of the strongest predictors of oral health outcomes in general. Our study’s sample is homogenous with regard to being low-income because that is a qualification for enrollment at CrossOver. Yet, within-income-group oral health outcome stratification is observed at the population level by race, ethnicity, and other characteristics [1]. While we do not report in this paper on the distribution of race or ethnicity in our study sample, the poverty rate among Black and Hispanic residents of Richmond Virginia, where the study took place, is more than twice the poverty rate among white residents [32]. At the same time, Richmond’s population of residents born outside of the U.S. is almost half that of the United States proportion (7.03% versus 13.7%, [33]). Thus, U.S.-born health care safety net patients in Richmond Virginia may be disproportionately more exposed throughout their lives to a variety of factors that harm oral health such as racially biased exclusions from dental care or sub-optimal treatment decisions [34,35], political determinants of health such as redlining and inadequate access to healthful foods [36], or commercial determinants such as excess exposure to commercial tobacco outlets [37]. In other words, in contrast to the historically advantaged index groups (e.g., white, middle class) to whom minoritized groups are compared in population-level oral health studies (e.g., [24,26,28,29]), many or most participants enrolled in our study may have experienced deleterious conditions that negatively affect their oral health above and beyond income constraints, including but not limited to geographic and schedule barriers to care, structural ethno-racism and xenophobia, and dental benefits absence from Virginia’s adult Medicaid benefit prior to 2020. We will explore these topics when analyzing qualitative data collected for this study.

This study has a number of limitations. We analyzed data from a small sample of individuals who obtain health care at one FQHC in Richmond, Virginia. Results may not be generalizable to other groups of people. During data collection, a number of local and global sociopolitical shifts occurred, all of which could have informed participant enrollment, clinical and utilization measures, and self-report, including: the expansion of adult health care coverage in Virginia in 2019 under the provisions of the Affordable Care Act and CrossOver Healthcare Ministry’s new acceptance of Medicaid under this expansion; CrossOver’s expansion of its dental employees to include one FTE dentist in addition to its cadre of staff hygienists and volunteer dentists; and the COVID-19 pandemic and its effects on health service delivery including temporary mandatory service limitations and the expansion of telehealth, neither of which are notated in EHR data. Reason for rescheduling was not recorded, including whether the reschedule was initiated by the patient or the clinic.

## 5. Conclusions

Oral health outcomes result from the accumulation of complex and interacting risk and protective factors across the lifespan that have consequences from childhood through advanced age. Among im/migrants to the U.S., differentiated oral health outcomes by age-at-arrival, including retention of natural dentition, dental appointment completion, and self-rated oral health and oral health related quality of life indicates the importance of investigating im/migration dynamics in oral health studies that use lifecourse theory, for example the lasting effects of preventive services, fluoridated water, and expectations of oral health while in communities of origin, and exposure to these factors plus iatrogenic corporatism (high-sugar foods, alcohol, tobacco) among the children of people who im/migrated at a young age.

## Figures and Tables

**Figure 1 ijerph-19-01477-f001:**
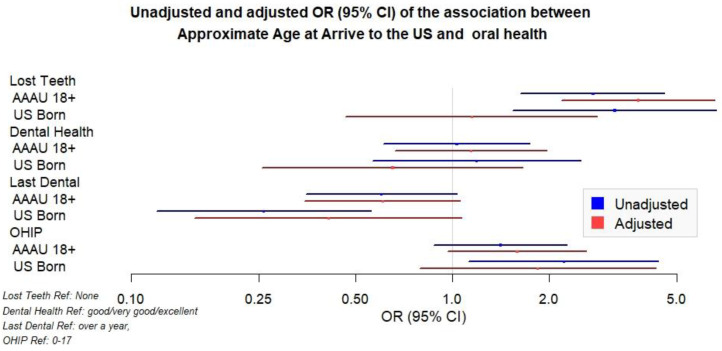
Unadjusted and adjusted OR (95% CI) of the association between approximate age at arrival to the U.S. and oral health outcomes.

**Figure 2 ijerph-19-01477-f002:**
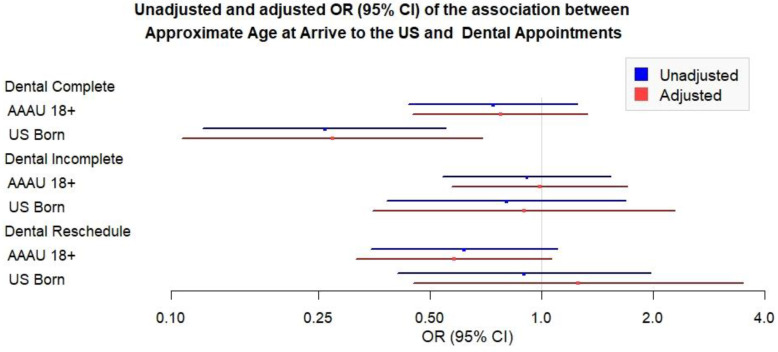
Unadjusted and adjusted OR (95% CI) of the association between approximate age at arrival to the U.S. and dental appointments.

**Table 1 ijerph-19-01477-t001:** Distribution of socio-demographic variables and how they differ according to age of arrival to US (AAAU).

	N (%)	Approximate Age at Arrival to U.S. N (%)			
Variable	Overall N = 327	U.S. Born (USB) N = 48	Age of Arrival <18 (AAAU < 18) N = 87	Age of Arrival ≥18 (AAAU ≥ 18) N = 189	*p*-Value
Age *	46 (36.5, 80)	50 (46, 60)	38 (35, 41)	50 (46, 60)	<0.0001
Gender					0.0070
Male	93 (28.6)	21 (43.8)	16 (18.4)	56 (29.8)	
Female	232 (71.4)	27 (56.3)	71 (81.6)	132 (70.2)	
Missing	2	0	0	1	
Language preference					<0.0001 **
English	61 (18.7)	47 (97.9)	2 (2.3)	11 (5.8)	
Spanish	220 (67.5)	1 (2.1)	74 (85.1)	145 (76.7)	
Other	45 (13.8)	0 (0.0)	11 (12.6)	33 (17.5)	
Missing	1	0	0	0	
Marital Status					0.0188
Married	123 (38.2)	10 (21.3)	31 (36.1)	81 (43.3)	
Not Married	199 (61.8)	37 (78.7)	55 (64.0)	106 (56.7)	
Missing	5	1	1	2	
Education Attainment					<0.0001
<High School	160 (50.0)	12 (25.0)	54 (65.9)	94 (50.0)	
High school/some college	106 (33.1)	29 (60.4)	22 (26.8)	54 (28.7)	
college or more	54 (16.9)	7 (14.6)	6 (7.3)	40 (21.3)	
Missing	7	0	5	1	
Alcohol Use					0.0081
No	220 (68.1)	24 (50.0)	57 (66.3)	137 (73.3)	
Yes	103 (31.9)	24 (50.0)	29 (33.7)	50 (26.7)	
Missing	4	0	1	2	
Smoking Status					<0.0001
Never	210 (67.7)	18 (39.1)	62 (76.5)	129 (70.9)	
Former	71 (22.9)	14 (30.4)	15 (18.5)	42 (23.1)	
Current	29 (9.4)	14 (30.4)	4 (4.9)	11 (6.0)	
Missing	17	2	6	7	
Income					0.0749
0–10,000	68 (28.6)	16 (39.0)	12 (18.8)	40 (30.5)	
10,000–20,000	99 (41.6)	13 (31.7)	35 (54.7)	49 (37.4)	
20,000+	71 (29.8)	12 (29.3)	17 (26.6)	42 (32.1)	
Missing	89	7	23	58	
Diabetes					0.3846
No	237 (73.6)	32 (66.7)	66 (77.7)	138 (73.8)	
Yes	85 (26.4)	16 (33.3)	19 (22.4)	49 (26.2)	
Missing	5	0	2		
Years in US					<0.0001 **
0–5 Years	58 (17.9)	0 (0.0)	1 (1.2)	57 (30.2)	
5–10 years	50 (15.4)	0 (0.0)	4 (4.6)	46 (24.3)	
10+ years	169 (52.0)	0 (0.0)	82 (94.3)	86 (45.5)	
Born in US	48 (14.8)	48 (100.0)	0 (0.0)	0 (0.0)	
Missing	2	0	0	0	

* Median/interquartile range and Kruskal–Wallis Test. ** Fisher’s Exact Test.

**Table 2 ijerph-19-01477-t002:** Self-reported oral health measures by approximate age at arrival to the U.S.

	N (%)	Approximate Age at Arrival N (%)			
Variable	Overall N = 327	USB N = 48	AAAU < 18 N = 87	AAAU > 18 N = 189	*p*-Value
Lost teeth					0.0016
None	80 (25.7)	7 (15.9)	34 (40.5)	39 (21.3)	
1–8 teeth	178 (57.2)	28 (63.6)	44 (52.4)	106 (57.9)	
8+ teeth	53 (17.0)	9 (20.5)	6 (7.1)	38 (20.8)	
Dental health					0.8934
Poor/Fair	181 (58.2)	27 (61.4)	48 (57.1)	106 (57.9)	
Good/Very good/ Excellent	130 (41.8)	17 (38.6)	36 (42.9)	77 (42.1)	
Last Dental Visit					0.0020
Within past year	171 (55.0)	15 (34.1)	56 (66.7)	100 (54.6)	
Over a year	140 (45.0)	29 (65.9)	28 (33.3)	83 (45.4)	
OHIP *	21.0 (16.0, 31.0)	24.5 (18.0, 35.5)	19.0 (15.0, 26.5)	22.0 (16.0, 32.0)	0.0306

* Median/interquartile range and Kruskal–Wallis Test for continuous measures. Lost teeth, Dental Health, last dental visit, and OHIP: N = 311.

**Table 3 ijerph-19-01477-t003:** Self-reported health care appointment outcomes by approximate age at arrival to the U.S.

	N (%)	Approximate Age at Arrival N (%)			
Variable	Overall N = 327	USB N = 48	AAAU < 18 N = 87	AAAU > 18 N = 189	*p*-Value
**Dental Appointments**					
Dental incomplete					0.8446
None	202 (62.9)	31 (66.0)	53 (60.9)	118 (63.1)	
At Least Once	119 (37.1)	16 (34.0)	34 (39.1)	69 (36.9)	
Dental Complete					0.0011
None	143 (44.6)	32 (68.1)	31 (35.6)	80 (42.8)	
At Least Once	178 (55.5)	15 (31.9)	56 (64.4)	107 (57.2)	
Dental reschedule					0.2265
None	243 (75.7)	34 (72.3)	61 (70.1)	148 (79.1)	
At Least Once	78 (24.3)	13 (27.7)	26 (29.9)	39 (20.9)	
**Medical Appointments**					
Medical Incomplete					0.6727
None	40 (12.7)	5 (11.1)	13 (15.5)	22 (11.9)	
At Least Once	274 (87.3)	40 (88.9)	71 (84.5)	163 (88.1)	
Medical complete					0.0212 **
None	5 (1.6)	3 (6.7)	1 (1.2)	1 (0.5)	
At Least Once	309 (98.4)	42 (93.3)	83 (98.8)	184 (99.5)	
Medical Reschedule					0.2435
None	206 (65.6)	30 (66.7)	61 (72.6)	115 (62.1)	
At Least Once	108 (34.4)	15 (33.3)	23 (27.4)	70 (37.8)	
**Mental Health Appointments**					
Mental Health incomplete					0.0065
None	236 (73.5)	27 (57.5)	72 (82.8)	137 (73.3)	
At Least Once	85 (26.5)	20 (42.6)	15 (17.2)	50 (26.7)	
Mental Health Complete					0.0258
None	220 (68.5)	25 (53.2)	66 (75.9)	129 (69.0)	
At Least Once	101 (31.5)	22 (46.8)	21 (24.1)	58 (31.0)	
Mental Health reschedule					0.0024
None	270 (84.1)	32 (68.1)	79 (90.8)	159 (85.0)	
At Least Once	51 (15.9)	15 (31.9)	8 (9.2)	28 (15.0)	

Dental Appointment: N = 321, Medical Appointment: N = 314, Mental Health Appointment: N = 321, ** fisher’s exact test. Blod: Distinguish headings

**Table 4 ijerph-19-01477-t004:** Dental appointment status by type of appointment according to approximate age at arrival to U.S.

	Approximate Age at Arrival to US (AAAU) N (%)		
Variable	USB N = 47	AAAU < 18 N = 87	AAAU ≥ 18 N = 187
Dental Complete-At least once	A = 28	A = 92	A = 189
Pain	4 (14.3)	10 (10.9)	35 (18.5)
Preventative	11 (39.3)	50 (54.3)	93 (49.2)
Restoration	9 (32.1)	20 (21.7)	37 (19.6)
Extractions	4 (14.3)	12 (13.0)	24 (12.7)
Dental Incomplete-At least once	A = 27	A = 48	A = 83
Pain	3 (11.1)	2 (4.2)	5 (6.0)
Preventative	15 (55.6)	20 (41.7)	50 (60.2)
Restoration	7 (25.9)	18 (37.5)	15 (18.1)
Extractions	2 (7.4)	8 (16.7)	13 (15.7)
Dental reschedule-At least once	A = 13	A = 29	A = 43
Pain	0 (0.0)	1 (3.4)	6 (14.0)
Preventative	7 (53.8)	20 (69.0)	22 (51.2)
Restoration	5 (38.5)	4 (13.8)	10 (23.3)
Extractions	1 (7.7)	4 (13.8)	5 (11.6)

Only those with complete dental data were included in this table. A: is the total number of appointment types completed per age at migration status and appointment status. Note, an individual could have several different types appointments in each group or none. Dental incomplete includes no-shows and cancellations. Pain: include pain evaluations, all appointments that include pain, prescriptions, and Abscess. Preventative: scaling/Curettage/root planting, Cleaning, dental exams/evals, consultations, diabetic appointments, oral hygiene education, Enameloplasty Sealant Technique, evaluations on dentures/fillings/lesions/sores, fixing dentures, and all non-specified appointments performed by a dental hygienist. Restoration: Fillings, restorations, bridge falling off, Endodontic, recementing crowns, upper denture. Extractions: extractions, broken/cracked tooth, dry sockets, and possible extractions.

## Data Availability

The data presented in this study are available on request from the corresponding author. The data are not yet publicly available due to ongoing analyses and reporting by the study team.

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
