# Peer review of "Oral Health, Health Service Utilization, and Age at Arrival to the U.S. among Safety Net Patients"

_ijerph, 2022, doi:10.3390/ijerph19031477_

Round 1

Reviewer 1 Report

I would like to congratulate the authors on their manuscript which aims to investigate immigrant1s oral health using a lifecourse approach. I am not a native English speaker; however, I suggest an minor spell checking of the manuscript before submitting. (in. 3.2.1. 5th row „likel”, or sometimes life course or lifecourse and more...)

The introduction part is disproportionately long and the two last paragraphs of introduction would be negotiated in the discussion section.

Please define a transgenerational economic mobility.

When writing the manuscript perhaps the most important part to get right is the materials and methods section. This section is usually divided in subsections, the firs of which should cover the study population (where the interventions took place, ethical approval data and statement, inclusion and exclusion criteria).

There are 3 other methods- they completed the questionnaire on their own either with help of assistant or via telecommunication. It seems the weakest point of investigation. In which methods the subjects can appropriate answer? There could be significant different between these answers.

Why is it important the bilingual assistant? Are they standardized or only assistant read the questionnaire?

Please discuss the validation process of self-reported oral health questionnaire, the questionnaire should be displayed as an appendix.

The 3 subgroups are inhomogeneous by gender and age,  the authors would discuss the comparing of 3 subgroups.

In table 2 the columns and rows are very unclear, it would be great separate tables (i: lost teeth, dental health last dental visit and OHIP ,ii: dental appointment, iii: medical appointment and iv: mental health appointment. (subjects are different and the authors should discuss the exclusion criteria).

Table 3 would be modify according pain, preventative, restoration, extrations- and subgroups dental complete/incomplete/dental reschedule at least once.

In the discussion section more hypothesis are not discussed or not correct concluded in the AAAU18 and USB groups.

Author Response

Dear Editors and Reviewers,

Thank you for your thoughtful review of our paper. We appreciate your close reading and suggestions.

We are addressing both reviewers’ responses together in one document because some of the feedback offered by Reviewer 1 and Reviewer 2 conflicts with the other. Therefore, it is important that we address this feedback in a forum where both reviewers can consider all responses to all reviewer feedback.

We look forward to hearing next steps in this process.

--Sarah Raskin, on behalf of all authors

Reviewer 1

  1. I would like to congratulate the authors on their manuscript which aims to investigate immigrant1s oral health using a lifecourse approach. I am not a native English speaker; however, I suggest an minor spell checking of the manuscript before submitting. (in. 3.2.1. 5th row „likel”, or sometimes life course or lifecourse and more...)

AUTHOR RESPONSE: Thank you for calling our attention to this matter. We did not realize that the author template did not automatically identify spelling and grammatical errors. We have run spellcheck and grammar check on the revision.

  1. The introduction part is disproportionately long and the two last paragraphs of introduction would be negotiated in the discussion section.

AUTHOR RESPONSE:

  • Given the paper topic, the special issue to which the paper is being submitted, and the data we analyze, we found it important to contextualize the paper within the following knowledge domains: Oral disease etiology (Para 1); Lifecourse theories of oral health (Paras 2 and 3) and gaps therein (Para 3); Immigration as a social determinant of health and lifecourse theory (Para 4); and Immigrants’ oral health (Para 5). An important and novel contribution of our paper is to foster conversations among these disparate knowledge domains, which is best done by giving each knowledge domain its due. The length of the introduction is consistent with innumerable papers featured in IJERPH’s Editor’s Choice Collection and its Most Cited and Viewed Collection, and is shorter than that of many papers featured in those collections.
  • We interpret the author’s comment re: “negotiated in the discussion section” to indicate the importance of discussing the findings in light of the paragraphs on immigration and lifecourse theory (Intro Para 4) and immigrants’ oral health outcomes (Intro Para 5). The entire pretext of our findings and subsequent discussions is grounded in these topics. For example:
    • The association between age-at-migration and health outcomes (Intro Para 4 and Discussion Paras 1, 2, 3 and 5); In particular, in Discussion Para 2 we theorize hypotheses for AAAU<18s risk and protective exposures as a response to the reconceptualization of age-at-migration as a set of dynamic lifecourse circumstances rather than measure of acculturation (per Intro Para 4).
    • Immigrants’ oral health outcomes, with an emphasis on their disproportionate burden of untreated oral diseases (Intro Para 5 and Discussion Paras 2, 3, 4, and 5).
  • In the Discussion section, we have added a new paragraph (Discussion Para 6, page 11) that addresses our findings regarding superior oral health outcomes of non-US-born participants who arrived prior to age 18, as compared with US-born participants. We posit hypotheses that should be explored in future research. These findings contradict existing knowledge. We are grateful that the reviewer’s comments prompted us to consider these findings with greater attention.

  1. Please define a transgenerational economic mobility.

AUTHOR RESPONSE: Completed. Please see page 2, where we added the likelihood of children’s lifetime economic stability exceeding their parents’

  1. When writing the manuscript perhaps the most important part to get right is the materials and methods section. This section is usually divided in subsections, the firs of which should cover the study population (where the interventions took place, ethical approval data and statement, inclusion and exclusion criteria).

AUTHOR RESPONSE: We prepared the manuscript in adherence with STROBE guidelines for observational studies. Please note:

  • This is an observational study, not an intervention study. No intervention occurred.
  • We refer to the location of recruitment and data collection on page 2, in Materials and Methods Para 1: CrossOver Healthcare Ministry, a federally qualified health center (FQHC) in Richmond, Virginia
  • The ethical approval is provided on page 3, in Methods and Materials Section 2.2 Data Source: This study was approved by the Virginia Commonwealth University (VCU) Institutional Review Board (IRB), Protocol #HM20014861. We do not commonly report the date of approval but have added it at the reviewer’s request.
  • Eligibility criteria was already provided on page 3, in Methods and Materials Section 2.1 Setting, Study Population, and Participants. We clarified Inclusion and Exclusion criteria in this section.

  1. There are 3 other methods- they completed the questionnaire on their own either with help of assistant or via telecommunication. It seems the weakest point of investigation. In which methods the subjects can appropriate answer? There could be significant different between these answers.

AUTHOR RESPONSE: We understand that a gold standard of data collection is consistency across all participants when possible, for example with a sample that is homogenous with regard to literacy. At the same time, the reality of primary data collection means that sometimes studies have to adjust to real life circumstances. We work in the established post-positivist tradition of observational community engaged research that embraces real world dental experiences and dental service delivery rather than the control of factors for uniformity (e.g. inclusion/exclusion criteria that are overly rigid to pre-control analytic possibilities).

Importantly, literacy in order to complete a paper form was not an inclusion criterion prior to the data collection transitions necessitated by COVID discussed below. As community engaged research initiated by our partners at the safety net clinic CrossOver, we aimed to invite participation from as large and diverse an array of patients as possible to ensure our findings’ salience to their practice and validity with regard to representing patients’ perspectives. To have limited enrollment only to individuals prepared to independently read and notate answer options would have been to exclude a substantial swath of their patient population; the inverse is true as well, with regard to individuals who preferred to have the forms read and provide responses. The research assistant who collected participants’ responses read the questions and answer options verbatim, and rehearsed reading them with a neutral tone so as to minimize their influence on participant responses. The forms that were collected using this approach were not distinguished from the forms that participants completed by hand, so as to not influence analysis. Both were dual-entered by other research assistants to assure high quality and error-checking opportunities in data entry.

In March 2020, the VCU IRB required that we—like nearly all studies worldwide—shut down data collection in response to the COVID-19 pandemic, as a safety precaution. We felt an exceptional responsibility to temporarily halt data collection due to the excess vulnerability to infection of safety net patients due to risk exposures such as employment in low-wage hourly positions. Moreover, in March 2020 our community partner **rightly** prohibited our in-person presence for the protection of its patients and staff, and our staff. After numerous conversations with our project team, community partners, and the IRB, we proposed to resume data collection in August 2020 using virtual data collection. We determined that having trained research assistants read the form via phone would be the best way to continue with the norm of enrolling as many and as diverse participants as possible. We were concerned that the digital divide would be a barrier to the high quality and thorough completion of collecting data using secure online forms or videoconferencing, and we did not feel confident in the response rate or confidentiality of mailed forms. Therefore, we found collecting data via telephone calls to be the most suitable way to complete this study.

Importantly, we were determined to complete this study, even under suboptimal recruitment, enrollment, and data collection circumstances, as a good faith gesture toward our community partner. We sincerely hope that this effort is considered appreciable by reviewers rather than a methodological indictment, as it reflects community engaged research values.

  1. Why is it important the bilingual assistant? Are they standardized or only assistant read the questionnaire?

AUTHOR RESPONSE: As stated in Methods section 2.1, 75% of CrossOver’s patient population identifies a language other than English as their first language (of whom 80% are Spanish primary or monolingual). Enrolling a diverse participant population with regard to migration experience was foundational to the research questions that guide this study. Therefore, it was essential that our research assistants be bilingual. Our four bilingual research assistants during the data collection and management phase had different responsibilities, which were supervised and calibrated by the PI. These included collecting data both in person and digitally, and double-entering data (for quality control) from forms into the REDCap database.

  1. Please discuss the validation process of self-reported oral health questionnaire, the questionnaire should be displayed as an appendix.

AUTHOR RESPONSE: We selected questions on the self-reported oral health instrument from existing instruments that had been validated by other scholars. They reflected questions not covered elsewhere in the validated instruments that we administered, or the electronic health record data that we were permitted by our community partner to analyze.

  1. The 3 subgroups are inhomogeneous by gender and age, the authors would discuss the comparing of 3 subgroups.

AUTHOR RESPONSE: We briefly discuss participant demographics in Results section 3.1.1. Socio-demographic data. We discuss the potential role of age with regard to outcome variables in Discussion Para 2: Study participants who arrived to the U.S. prior to age 18 had better retention of natural dentition than those who arrived after age 18 or were born in the U.S. This finding may reflect the study population’s on-average younger median age (38 years old) as compared with their counterparts’ median age (50 years old).

  1. In table 2 the columns and rows are very unclear, it would be great separate tables (i: lost teeth, dental health last dental visit and OHIP ,ii: dental appointment, iii: medical appointment and iv: mental health appointment. (subjects are different and the authors should discuss the exclusion criteria).

AUTHOR RESPONSE: Thank you for your comment. We have now split Table 2 into a Table 2a (oral health measures) and a Table 2b (health care appointments-dental, medical and mental health). Please note that while the sample sizes differ based on whether it's an oral health measure or health care appointment, the study population is the same, and the difference in sample sizes pertains to whether a given participant completed a healthcare appointment or reported an oral health measure.

  1. Table 3 would be modify according pain, preventative, restoration, extrations- and subgroups dental complete/incomplete/dental reschedule at least once.

AUTHOR RESPONSE: We agree that it would be important to assess modification according to pain. However, we have no indication of pain for the different appointment types (preventative, restorative, extractions). The pain specified in Table 3 and under the methods section (2.3.2) pertains to soft tissue pain including abscesses.

  1. In the discussion section more hypothesis are not discussed or not correct concluded in the AAAU18 and USB groups.

AUTHOR RESPONSE: Please see Discussion Para 6  (discussed above )regarding new hypotheses of USB’s comparatively poor oral health outcomes. Hypotheses are also included in existing Discussion, for example (Para 2) the roles of sugar intake, microbiome composition, and metabolic disorders (as documented in other lifecourse studies of immigrants’ health); (Para 3) the role of expectations of changing dentition over the lifecourse; and (Para 5) the role of opportunities to complete long-delayed care.

Reviewer 2 Report

The subject of the paper is extremely interesting. Well, it combines a study of (dental) medicine with the social and economic variables of the population. As well as the relationship between both realities (immigration, ages, etc.).

Anyway, I detect several weaknesses. Although 327 patients are a significant number, as a statistical basis it is not that large. But if it would be enough if all that information was better used. It gives me the feeling that it is a first investigation and, although interesting, it is very descriptive. To be published in this journal, a much stronger and more innovative statistical treatment must be carried out. Do not make just a few tables with the data and a weak treatment. They should think of a much better and different statistical treatment. On the other hand, it would be good to describe or analyze the geographic characteristics of the place where the data was collected. Talk about the political, economic and social characteristics of Virginia. As well as delving into the origin of immigrants. I think the database and the theme are very interesting. But to be published, they must go a long way in their analysis and not just describe and cite the data.

Author Response

Dear Editors and Reviewers,

Thank you for your thoughtful review of our paper. We appreciate your close reading and suggestions.

We are addressing both reviewers’ responses together in one document because some of the feedback offered by Reviewer 1 and Reviewer 2 conflicts with the other. Therefore, it is important that we address this feedback in a forum where both reviewers can consider all responses to all reviewer feedback.

We look forward to hearing next steps in this process.

--Sarah Raskin, on behalf of all authors

Reviewer 2

  1. The subject of the paper is extremely interesting. Well, it combines a study of (dental) medicine with the social and economic variables of the population. As well as the relationship between both realities (immigration, ages, etc.). Anyway, I detect several weaknesses. Although 327 patients are a significant number, as a statistical basis it is not that large. But if it would be enough if all that information was better used. It gives me the feeling that it is a first investigation and, although interesting, it is very descriptive.

AUTHOR RESPONSE: The Reviewer is correct that this is formative/pilot community-engaged research. The n is consistent with similar studies, including many published in IJERPH.

  1. To be published in this journal, a much stronger and more innovative statistical treatment must be carried out. Do not make just a few tables with the data and a weak treatment. They should think of a much better and different statistical treatment.

AUTHOR RESPONSE: We appreciate the intention of this comment but we disagree with it.

  • IJERPH publishes many papers that are purely descriptive, including in the Editor’s Choice Collection and the Most Cited and Viewed Collection. We have strengthened our Discussion section with more interpretation of our findings.
  • The innovation in our analyses is in considering age at arrival to the U.S. as a critical moment in a lifecourse process rather than its established—and contested—use as a proxy for acculturation. By testing this independent variable, we are able to produce hypotheses about processual risk and protective factors to explain the outcomes identified, in particular AAAU<18s oral health outcomes that are favorable as compared with USBs and AAAU>18s. We look forward to examining these hypotheses by analyzing the qualitative data that complements our quantitative dataset, as well as continuing on with our statistical analyses and encouraging others to do the same.
  • The overly broad adjectives used in the reviewer’s comment makes it hard for us to respond beyond the above. For example, what “weak treatment” do they critique (please see our Discussion section)? Do they have a specific statistical analysis in mind?

  1. On the other hand, it would be good to describe or analyze the geographic characteristics of the place where the data was collected. Talk about the political, economic and social characteristics of Virginia. As well as delving into the origin of immigrants.

AUTHOR RESPONSE: We agree that the political, economic, and social characteristics of Virginia are relevant, generally, for understanding all safety net patients’ oral health. At the same time, the reviewers’ suggestions are beyond the scope of this paper.

  • We briefly review political, economic, and social determinants of disease in Discussion Paragraph 6.
  • We are reluctant to make the Introduction lengthier, as its length was identified as excessive by Reviewer 1 (see above).
  • We did not systematically collect year of migration, country of origin, or other factors that would make it more feasible to conduct statistical analyses indicated by this comment. Moreover, these are beyond the scope of the research question in this paper.
  • We will be better able to analyze these topics in our analysis of the qualitative data that correspond with the quantitative data analyzed in this paper.

Round 2

Reviewer 1 Report

I accept in present form. I suggest inserting the supplementary files or the OR should discuss in the discussion section.

Author Response

Thank you for drawing our attention to this oversight. We meant to include it in the first revision. We have included original survey questions as Appendix A.